



# The effect of spin polarization in DEER spectroscopy

Sarah R. Sweger[1], Vasyl P. Denysenkov[2], Lutz Maibaum[1], Thomas F. Prisner[2], and Stefan Stoll[1]

[1]Department of Chemistry, University of Washington, Seattle, WA 98195, USA
[2]Institute of Physical and Theoretical Chemistry and Center of Biomolecular Magnetic Resonance, Goethe University Frankfurt am Main, Frankfurt, Germany

**Correspondence:** Stefan Stoll (stst@uw.edu)

**Abstract.** Double electron–electron resonance (DEER) spectroscopy measures the distribution of distances between two electron spins in the nanometer range, often on doubly spin-labeled proteins, via the modulation of a refocused spin echo by the dipolar interaction between the spins. DEER is commonly conducted under conditions where the polarization of the spins is small. Here, we examine the DEER signal under conditions of high spin polarization, thermally obtainable at low temperatures

and high magnetic fields, and show that the signal acquires a polarization-dependent out-of-phase component, both for the intra- and the inter-molecular contribution. For the latter, this corresponds to a phase shift of the spin echo that is linear in the pump pulse position. We derive a compact analytical form of this phase shift and show experimental measurements using monoradical and biradical nitroxides at several fields and temperatures. The effect highlights a novel aspect of the fundamental spin physics underlying DEER spectroscopy.

## 1 Introduction

Double Electron–Electron Resonance (DEER) spectroscopy is a pulse Electron Paramagnetic Resonance (EPR) technique utilized for determining distances between spin centers on a nanometer scale (Milov et al., 1981; Larsen and Singel, 1993; Milov and Tsvetkov, 1997). The technique has seen use with studies of organic polymers but is most commonly used for structural studies in large biomolecules, such as proteins and nucleic acids (Schiemann and Prisner, 2007; Tavenor et al., 2014;

Duss et al., 2014; Manglik et al., 2015; Verhalen et al., 2017; Barth et al., 2018; Dastvan et al., 2019; Evans et al., 2020; Born et al., 2021). DEER resolves the full distribution of distances in an ensemble of proteins, making it possible to directly quantify protein conformational ensembles (Larsen and Singel, 1993; Jeschke and Polyhach, 2007; Jeschke, 2012).

DEER measures the amplitude of a spin echo as a function of the position of one or more pump pulses. The magnetic dipole–dipole coupling between spins leads to a modulation of the echo amplitude whose frequency is dependent on the magnitude

of the coupling. The majority of DEER experiments are conducted on samples of dilute and uniformly distributed spin pairs on doubly-labeled molecules or complexes. These samples give rise to an overall signal that is a product of an oscillatory intra-molecular signal and an exponentially decaying inter-molecular signal (Fig. 1A). The latter is often referred to as the background signal.

The behavior of the spins depends upon temperature and magnetic field, which together determine the magnitude of thermal

polarization (Fig. 1B). DEER spectroscopy using nitroxide radicals is often conducted at 50-70 K and 0.3-1.2 T. Under





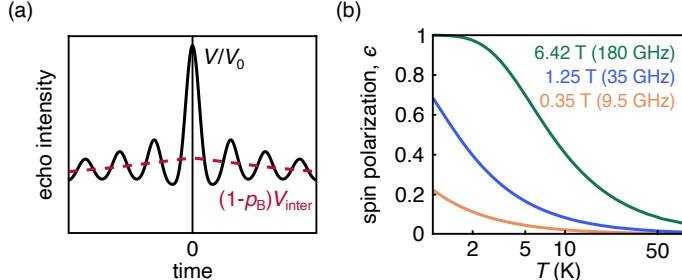

**Figure 1.** A. DEER signal (black) with the oscillatory intramolecular signal; the contribution from the exponentially decaying intermolecular signal is shown in pink. B. Dependence of thermal spin polarization $\epsilon$ on magnetic field and temperature.

these experimental conditions, the Zeeman interactions are smaller than the thermal energy, and consequently the thermal spin polarization is small. Larger-than-thermal polarization can be generated utilizing photo-excited molecular triplet states (Di Valentin et al., 2016; Dal Farra et al., 2020) or by optical pumping in NV centers, which have a triplet ground state (Takahashi et al., 2008; Stepanov and Takahashi, 2016; Sushkov et al., 2014; Shi et al., 2015). Non-thermal spin polarizations are known to lead to out-of-phase electron spin echo amplitude modulation (OOP-ESEEM) in spin-correlated radical pairs (Salikhov et al., 1992; Tang et al., 1994).

This work considers DEER signals under significant thermal spin polarization, as obtainable at low temperatures and high magnetic fields. Expanding on previous work(Marko et al., 2013), we show theoretically that the out-of-phase components of both the intra- and inter-molecular signals are polarization-dependent. In particular, we derive a closed-form analytical expression for the inter-molecular signal that reveals a polarization-dependent phase factor. We demonstrate this experimentally with monoradical and biradical nitroxide samples at several fields and temperatures.

The paper is structured as follows. Section 2 steps through the analytical spin dynamics model showing the polarization-dependent echo phase, beginning from a two-spin system and then expanding to a uniform spatial distribution of spins. Section 3 describes experimental details, and Section 4 presents and discusses the experimental results. Section 5 concludes.

## 2 Analytical Model

In this section, we derive the analytical form of the inter-molecular DEER signal, without the usual assumption of the high-temperature limit. The derivation within the high-temperature limit and the initial analytical expression for the intra-molecular DEER signal of an isolated spin pair beyond the high-temperature limit have previously been reported (Milov et al., 1973; Marko et al., 2013; Dal Farra et al., 2019) and is derived in Section S1 of the Supporting Information.

The analysis is based on the standard model for 4-pulse DEER of a frozen dilute solution of doubly spin-labeled molecules. We define two spectrally non-overlapping and separately addressable subsets of electron spins, A and B. (The sample also contain spins not affected by any pulses due to the broad EPR spectrum, particularly at high fields; we denote them as C spins.) The A-spins are manipulated via pulses at the probe frequency, $\omega_A$, while the B-spins are inverted via a pulse at the pump





frequency, $\omega_{\mathrm{B}}$. DEER measures the amplitude $V(t)$ of the refocused electron spin echo as a function of the position in time of
the pump pulse, $t$. It is given by

$$V(t) = V_0 \cdot V_{\mathrm{intra}}(t) \cdot V_{\mathrm{inter}}(t) \tag{1}$$

The echo amplitude is a product of an overall amplitude, $V_0$, a modulation function due to the intra-molecular spin pairs,
$V_{\mathrm{intra}}(t)$, and an inter-molecular modulation function, $V_{\mathrm{inter}}(t)$, containing the contributions from spin pairs with spins on
different molecules. For an individual pair of one A-spin located at position $\boldsymbol{r}_{\mathrm{A}}$ and one B-spin located at $\boldsymbol{r}_{\mathrm{B}}$ (not necessarily
on the same molecule), the DEER signal is (Marko et al., 2013)

$$V_{\mathrm{AB}}(\boldsymbol{r}, t) = (1 - p_{\mathrm{B}}) + p_{\mathrm{B}}\left(\cos(\omega t) + \mathrm{i}\epsilon \sin(\omega t)\right) \tag{2}$$

Here, $\boldsymbol{r} = \boldsymbol{r}_{\mathrm{B}} - \boldsymbol{r}_{\mathrm{A}}$ is the inter-spin distance vector, and $\omega$ is the dipolar coupling frequency

$$\omega(\boldsymbol{r}) = D\,\frac{1 - 3\cos^2\theta_{\mathrm{AB}}}{r^3} \qquad r = |\boldsymbol{r}| \qquad \cos\theta_{\mathrm{AB}} = \boldsymbol{r} \cdot \boldsymbol{z}/r \tag{3}$$

with the dipolar constant

$$D = \frac{\mu_0}{4\pi}\,\frac{\mu_{\mathrm{B}}^2 g_{\mathrm{e}}^2}{\hbar} \approx 2\pi \cdot 52.04\,\mathrm{MHz\,nm}^3 \tag{4}$$

where $\mu_0$ is the magnetic constant, $\mu_{\mathrm{B}}$ is the Bohr magneton, $g_{\mathrm{e}}$ is g-value of the free electron, $\hbar$ is the reduced Planck constant,
and $\boldsymbol{z}$ is the unit vector along the direction of the applied magnetic field. In Eq. (2), $p_{\mathrm{B}}$ is the inversion efficiency of the pump
pulse, and $\epsilon$ is the polarization of the B-spins, ranging between 0 and 1. For a spin–1/2, the polarization at thermal equilibrium
is

$$\epsilon(B, T) = \frac{N_\beta - N_\alpha}{N_\beta + N_\alpha} = \tanh\frac{g_{\mathrm{e}}\mu_{\mathrm{B}}B}{2k_{\mathrm{B}}T} \tag{5}$$

where $N_\beta$ and $N_\alpha$ are the populations of the ground and excited state, respectively, $B$ is the magnetic field strength, $k_{\mathrm{B}}$ is the
Boltzmann constant, and $T$ is the temperature. The dependence of the thermal polarization on $B$ and $T$ is shown in Fig. 1B.
Equation (2) is derived under the assumption that the dipolar coupling frequency is small compared to the difference in reso-
nance frequencies between the two spins. Equation (2) is the standard in-phase DEER signal with an additional polarization-
dependent out-of-phase sin term that is proportional to $\epsilon$ and therefore significant at low temperatures, high magnetic fields, or
in the presence of spin hyperpolarization.

Figure 2 uses a vector model in the rotating frame to visualize the dynamics of A-spins through the 4-pulse DEER sequence,
illustrating how the polarization affects the echo phase. The vectors indicate the two sub-ensembles of A-spins, one where the
neighboring B-spin is in the $m_S = +1/2$ state (blue) and the other where it is in the $m_S = -1/2$ state (pink). The lengths of
the arrows indicate the respective populations. The first two pulses of the sequence form an A-spin echo at time point 1, with
both A-spin sub-ensembles refocused with $-x$ phase. As time passes, the spins precess and dephase due to a distribution of
precession frequencies. This distribution is indicated by a set of arrows in varying shades, darker indicating higher frequency.



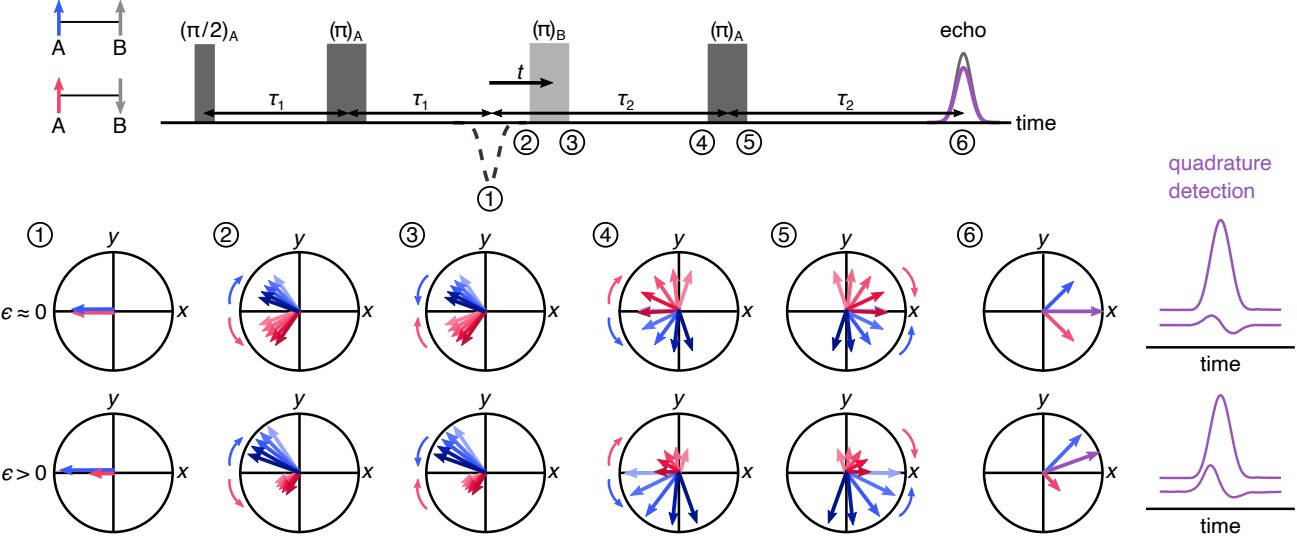

**Figure 2.** Vector model of the spin movement along the 4-pulse DEER sequence. Cases of low and high polarization values are shown where the numbers indicate the vector picture at a given point in time during the pulse sequence. The vectors indicate the two sub-populations of A-spins where those next to up-B-spins are colored blue and those next to down-B-spins are pink. The length of the arrows indicates the population. The overall magnetization is a sum of the two sub-populations, shown in purple. The respective echoes for low and high spin polarization values are shown on the right.

The mean precession frequencies of the two sub-ensembles differ by the dipolar coupling frequency $\omega$. The pump pulse flips the B-spins. This leaves all A-spin phases unaffected, but shifts the frequencies of the two A-spin sub-ensembles in opposite

directions by $\pm\omega$, so that the two sub-ensembles now accumulate phase in reversed directions, while continuing to dephase as before the pulse. The final $\pi$ pulse on the A-spins rotates both sub-ensembles around the $y$ axis, and after evolution for $\tau_2$, they refocus to give two echoes with opposite phases. The measured echo is proportional to the vector sum of these two sub-echoes. In the high-temperature regime ($\epsilon \approx 0$, top), the populations of the two sub-ensembles are equal, and therefore the echo has pure $x$ phase (in-phase), with the quadrature component ($y$ phase) integrating to zero. In the presence of significant

polarization ($\epsilon > 0$), the two sub-echoes have significantly different amplitudes, and the overall echo phase is rotated from $x$, yielding a non-zero quadrature (out-of-phase) component.

The average of $V_{\mathrm{AB}}$ over a uniform orientational distribution of A–B spin pairs with fixed $r$ is

$$V_{\mathrm{intra}}(r,t) = \frac{F_{\mathrm{C}}(z)\cos(\phi) + F_{\mathrm{S}}(z)\sin(\phi)}{z} + \mathrm{i}\epsilon \frac{F_{\mathrm{C}}(z)\sin(\phi) - F_{\mathrm{S}}(z)\cos(\phi)}{z} \tag{6}$$

with $\phi = Dt/r^3$ and $z = \sqrt{6\phi/\pi}$. $F_{\mathrm{C}}$ and $F_{\mathrm{S}}$ are the Fresnel cosine and sine integral functions respectively, which for imag-

inary arguments ($z$ is imaginary for $t < 0$) have the properties $F_{\mathrm{C}}(a\mathrm{i}) = \mathrm{i}\,F_{\mathrm{C}}(a)$ and $F_{\mathrm{S}}(a\mathrm{i}) = -\mathrm{i}\,F_{\mathrm{S}}(a)$. The in-phase (real) and out-of-phase (imaginary) components of this signal are shown in Fig. 3A.





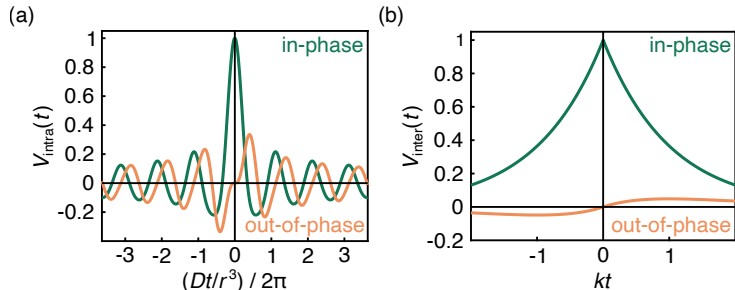

**Figure 3.** (**a**) The theoretical powder-averaged polarized DEER signal for a single A–B spin pair (Eq. (6)). (**b**) The theoretical DEER time trace for an A-spin coupled to a uniform three-dimensional distribution of B-spins at full spin polarization (Eq. (8), $\epsilon = 1$).

For a uniform, random, spatial distribution of B-spins in a sample, the signal is a product of all individual $V_{AB}$ signals starting from Eq. (2), additionally averaged over all B-spin configurations

$$V_{\text{inter}}(t) = \left\langle \prod_{b=1}^{N_B} V_{AB}(\boldsymbol{r}_b, t) \right\rangle \tag{7}$$

Here, $N_B$ is the number of B-spins in a configuration, and $\boldsymbol{r}_b$ is the vector from the A-spin to the $b$th B-spin. To arrive at the product form, the dipolar couplings among B spins are neglected. A somewhat involved derivation (spelled out in Section S2) shows that Eq. (7) evaluates to

$$V_{\text{inter}}(t) = \exp(-k|t|) \cdot \exp(i\alpha\epsilon kt) \tag{8}$$

where

$$k = \frac{8\pi^2}{9\sqrt{3}} p_B c_B D \qquad \alpha = \frac{\sqrt{3} + \ln(2 - \sqrt{3})}{\pi} \approx 0.13213 \tag{9}$$

The first factor in Eq. (8) is an exponential decay function, as has been derived and observed before. The decay rate constant $k$ depends on the B-spin concentration $c_B$ and on the inversion efficieny $p_B$. The second factor is an additional hitherto unappreciated phase factor with a phase that grows linearly with $t$ and is proportional to the spin polarization $\epsilon$. This phase factor leads to a non-zero out-of-phase signal for $t \neq 0$, as long as the spin polarization $\epsilon$ is large enough. The signal is plotted in Fig. 3B. Equation (8) was also confirmed numerically using Monte Carlo simulations (Fig. S1).

The extrema in the out-of-phase part of $V_{\text{inter}}(t)$ are located at

$$t = \pm t_{\text{OOP}} = \pm \frac{1}{k} \cdot \frac{\arctan(\alpha\epsilon)}{\alpha\epsilon} \approx \pm \frac{1}{k} \tag{10}$$

At these points, the magnitude of $V_{\text{inter}}$ has decayed to $e^{-1} \approx 0.368$. These time points depend only on concentration and inversion efficiency, not on polarization. Numerically, these quantities are related via

$$p_B \cdot (c_B/\text{mM}) \cdot (t_{\text{OOP}}/\mu\text{s}) \approx 1 \tag{11}$$



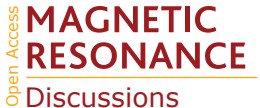

At $t = t_{\mathrm{OOP}}$, the out-of-phase amplitude of the inter-molecular signal is

$$\mathrm{Im}(V_{\mathrm{inter}}(t_{\mathrm{OOP}})) \approx \mathrm{e}^{-1} \frac{\alpha\epsilon}{\sqrt{1 + (\alpha\epsilon)^2}} \approx \mathrm{e}^{-1}\alpha\epsilon \approx 0.0486\,\epsilon \qquad (12)$$

At high temperatures and low fields, $\epsilon \approx 0$, and the out-of-phase component vanishes. The slope around $t = 0$ is dictated by the degree of polarization. Even at full polarization, the amplitude of the out-of-phase component is small (Fig. 3b), in contrast to

115 that of an isolated A–B pair (Fig. 3a), where the out-of-phase signal reaches $\approx 0.335\epsilon$ at $(Dt/r^3)/2\pi \approx 0.403$.

The Fourier transform of Eq. (8) is a Lorentzian lineshape

$$\sqrt{\frac{2}{\pi}} \frac{k}{k^2 + (\omega + \alpha\epsilon k)^2} \qquad (13)$$

centered at angular frequency $\omega = -\alpha\epsilon k$ and full width at half maximum of $fwhm = 2k$. The frequency $\omega$ is proportional to the polarization, the spin concentration, and the inversion efficiency. For $c_{\mathrm{B}} = 1\,\mathrm{mM}$, $p_{\mathrm{B}} = 1$, and $\epsilon = 1$, this gives a frequency

of $\omega/2\pi \approx -0.021\,\mathrm{MHz}$ and a line width of $fwhm/2\pi \approx 0.317\,\mathrm{MHz}$.

## 3 Materials & Methods

**Q-band DEER.** 2,2,6,6-Tetramethylpiperidine-1-oxyl (TEMPO) was obtained from Sigma-Aldrich and used to create an approximately 1 mM solution in 50:50 (w:w) $D_2O$ and $d_8$-glycerol. For measurements at Q-band, 30-50 $\mu$L of sample were syringed into 1.50 mm O.D. 1.1 mm I.D. quartz tubes (Sutter Instrument). Q-band experimental data were measured on a

125 Bruker Elexsys E580 spectrometer equipped with a Bruker D2 dielectric resonator at 33.9 GHz, 1.21 T, at 11 and 40 K. The microwave power was amplified with a 390 W Applied Systems Engineering TWT amplifier. The measurements were conducted at the field values that gave the maximum echo amplitudes. Optimal pulse lengths were determined using a Rabi nutation experiment. For the 11 K data, a standard 4-pulse DEER sequence was used with rectangular observer pulses at 33.842 GHz, the $\pi/2$ and $\pi$ pulses being 22 and 44 ns and a sech/tanh pump pulse applied at 33.922 GHz with a 200 ns length and 80

130 MHz bandwidth. For the 40 K data, rectangular observer pulses with 22 and 44 ns lengths were applied at 33.828 GHz and a sech/tanh pump pulse was applied at 33.908 GHz with a 200 ns length and 80 MHz bandwidth. Experiments at both temperatures utilized a 64-step phase cycle (Tait and Stoll, 2016). The values for $\tau_1$ and $\tau_2$ were 4000 and 4200 ns, respectively for the 11 K data and 3000 and 3200 ns for the 40 K data. To keep the sample at thermal equilibrium for each echo, the spectrum was collected with 1 shot per point and a 1 s repetition time at 11 K. At 40 K, the data was collected with 10 shots per point and a

135 3 ms repetition time.

**G-band DEER.** 4-Hydroxy-2,2,6,6-tetramethylpiperidine-1-oxyl (TEMPOL) was obtained from Sigma Aldrich Chemie GmbH and used to create an approximately 1.0 mM solution with 45:55 (v/v) $D_2O$:$d_8$-glycerol. The homo-biradical nitroxide was synthesized according to the procedure previously published (Bode et al., 2007), and 0.16 mg were dissolved in 0.61 mL of deuterated toluene, giving a final concentration of 0.25 mM, corresponding 0.5 mM total spin concentration. The solution

was placed in a 0.55 mm O.D. 0.4 mm I.D. quartz sample capillary that resulted in 250 nL of sample volume inside the TE011 cylindrical cavity resonator. Samples were frozen after insertion into the the cryostat. G-band experiment were performed on





a home-built pulse EPR spectrometer operating at 180 GHz, 6.42 T, and temperatures of 5, 40, and 50 K (Rohrer et al., 2001; Hertel et al., 2005). For the monoradical experiments, a standard 4-pulse DEER sequence with rectangular pulses was used with the observer $\pi/2$ and $\pi$ pulses being 36 and 58 ns, respectively, while the pump pulse was 58 ns with a frequency offset of +50 MHz. The values for $\tau_1$ and $\tau_2$ were 4000 and 5000 ns, respectively. Each DEER trace has 80 points and was recorded with 10 shots per point with 500 ms SRT for 5 K and 100 shots per point with 6 ms SRT for 40 K. This repetition time value was chosen as a 4-fold longer time with respect to $T_1$ value for nitroxide radicals in deuterated glycerol/$D_2O$ solution at 6.4 T and 40 K. For the experiments at 5 K, the repetition time was chosen according to the saturation recovery experiment (Fig. S5). Each trace was phased individually. For the biradical experiments, a standard 4-pulse DEER sequence with rectangular pulses was used with the observer $\pi/2$ and $\pi$ pulses being 48 and 90 ns, respectively, while the pump pulse was 80 ns with a frequency offset of +80 MHz. The values for $\tau_1$ and $\tau_2$ were 500 and 5000 ns, respectively. Each DEER trace has 80 points and was recorded with 10 shots per point with repetition time of 6 ms at 50 K, and 500 ms at 5 K.

## 4  Results & Discussion

To experimentally observe the polarization-dependent phase of the DEER signal, we performed DEER experiments on a 1 mM TEMPO sample in $D_2O$:$d_8$-glycerol at several magnetic fields and temperatures. The results are shown in Fig. 4. To fit the model from Eq. (8) to these data, we extended the model to

$$V(t) = V_0 \cdot \exp(-k|t - t_0|) \cdot \exp(\mathrm{i}\alpha\epsilon q_{\mathrm{B}}k(t - t_0)) \tag{14}$$

where the fit parameters are the overall signal amplitude $V_0$, the decay rate constant $k$, the zero-time shift $t_0$, and an additional phenomenological fit factor $q_{\mathrm{B}}$ discussed below. $\epsilon$ was calculated from temperature and magnetic field using Eq. (5) and is given in Fig. 4.

Figure 4a shows the DEER decay measured at 33.9 GHz, 1.21 T, and 11 K where the thermal polarization is about 7%. The out-of-phase signal is small but observable. The shape of the observed out-of-phase signal is reproduced by the fit. However, it grows with $t$ faster than predicted by theory. This is indicated by the fitted value $q_{\mathrm{B}} \approx 2.0(2)$, whereas we expect $q_{\mathrm{B}} = 1$ from theory (Eq. (8)). When the temperature is raised to 40 K, Figure 4b, thermal polarization is reduced significantly to 2%, and the out-of-phase signal flattens and is not observable as expected.

Figure 4c & d show the DEER decays measured at 180 GHz and 6.42 T, at 5 K and 40 K where the thermal polarization is 70% and 11%, respectively. Again, the model of Eq. (14) was fit to the data. In the 5 K data, the out-of-phase signal is now clearly visible, and the experiment again confirms the shape predicted by the model, with a discrepancy in the phase rotation rate ($q_{\mathrm{B}} \approx 3.6(2)$). There is also a very slight asymmetry in the in-phase signal that is not captured by the model. At 40 K, the amplitude of the out-of-phase signal is negligible and the in-phase signal is symmetric, consistent with the theoretical prediction.

The experimental results in Fig. 4 were obtained with a monoradical solution. To further investigate the origins of the observed deviation of the intermolecular contribution from the theoretical prediction, we measured a frozen solution of the

MAGNETIC
RESONANCE
Open Access Discussions

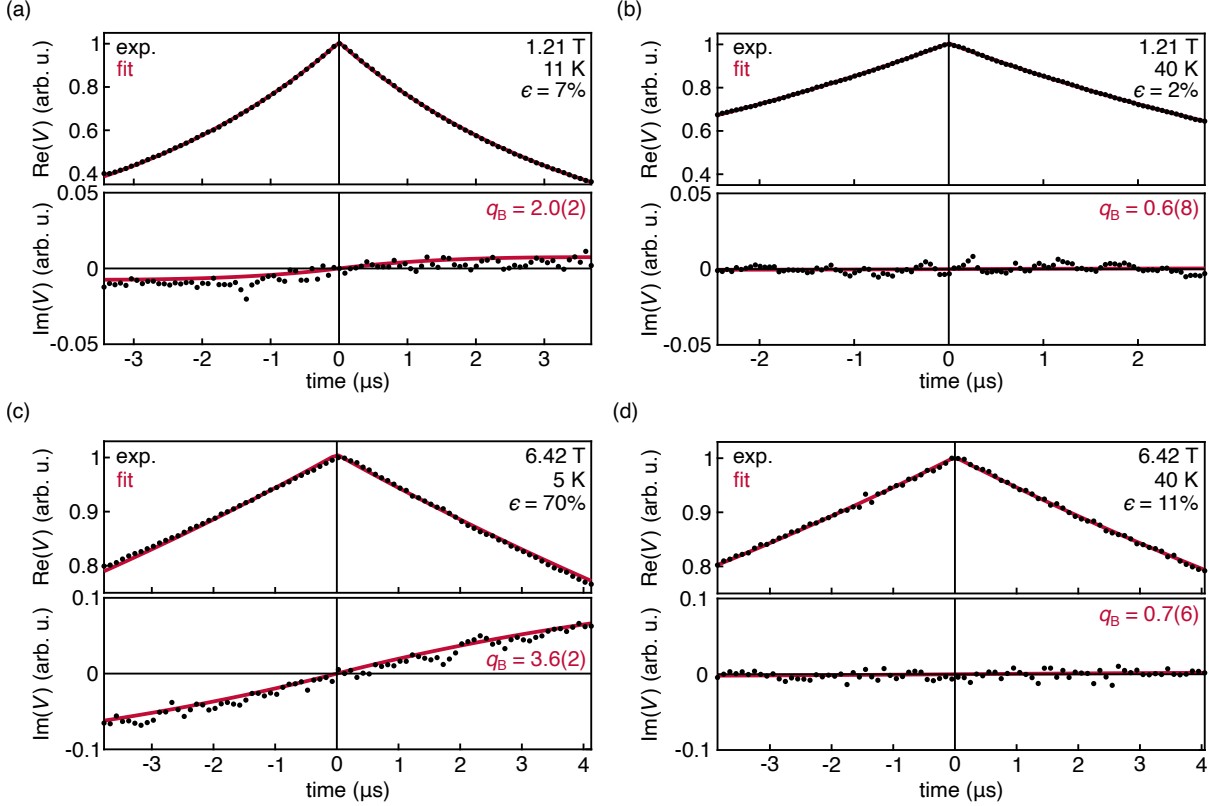

**Figure 4.** Experimental DEER traces for a 1.0 mM solution of TEMPO in 50:50 $D_2O$:$d_8$-glycerol, measured at 33.9 GHz at 11 K (a) and 40 K (b). Experimental DEER traces for a 1.0 mM solution of TEMPOL in 45:55 $D_2O$:$d_8$-glycerol, measured at 180 GHz and 6.42 T at 5 K (c) and 4d K (D). The data were fit with Eq. (14) and are shown with their 95% confidence interval in parentheses. All fit parameters are shown in Table S1.

nitroxide biradical shown in Fig. 5a at 180 GHz, 6.42 T, at 5 K and 50 K. Under these conditions, the thermal polarization

is 70% and 9%, respectively. The results are shown in Fig. 5. The intermolecular signal was fit according to Eq. (14) and the intramolecular signal was fit using an adaptation of Eq. (6) that incorporates an additional phenomenological factor $q_F$, analogous to $q_B$ in Eq. (14).

$$V_{AB}(r,t) = \frac{F_C(z)\cos(\phi) + F_S(z)\sin(\phi)}{z} + i\epsilon q_F \frac{F_C(z)\sin(\phi) - F_S(z)\cos(\phi)}{z} \qquad (15)$$

This was added to verify experimental temperature and to test whether the observed discrepancy in the intermolecular back-

ground signals ($q_B \neq 1$) is also present in the intramolecular signal. The intra-biradical distance distribution was modelled as a Gaussian distribution

$$P(r) = \frac{1}{w\sqrt{2\pi}} \exp\left(-\frac{(r-r_0)^2}{2w^2}\right) \qquad (16)$$

with the peak position $r_0$ and the standard deviation $w$ as fit parameters.

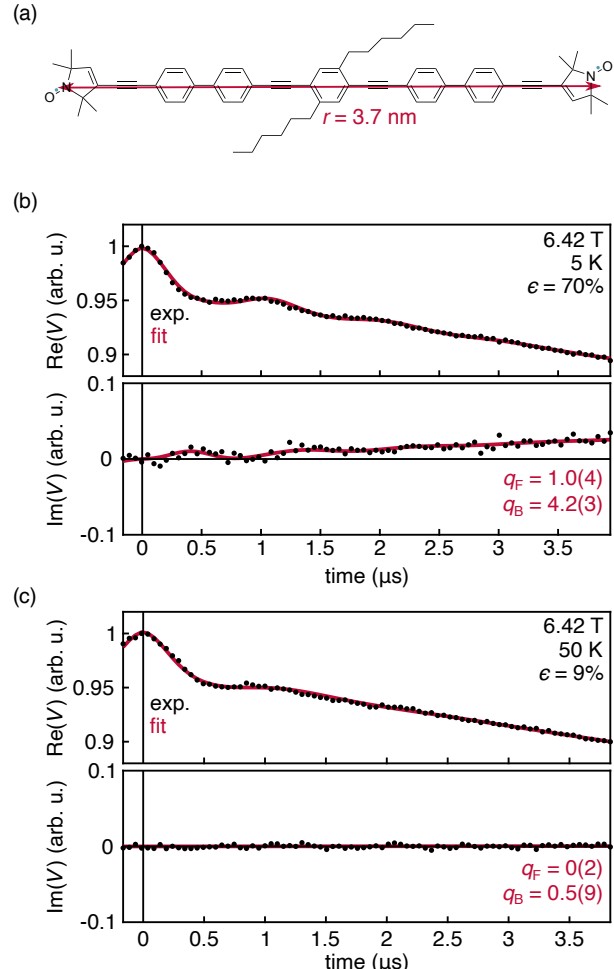

**Figure 5.** (a) Nitroxide biradical with an interspin distance of 3.7 nm (Schöps et al., 2015, 2016). Experimental DEER traces for a 0.25 mM solution of the biradical in deuterated toluene, measured at 180 GHz and 6.42 T at 5 K (b) and 50 K (c). The data was fit with Eqs. (14), (15), and (16). Fit parameters are shown with their 95% confidence interval. All fit parameters are listed in Table S2.

The experimental data are shown in Fig. 5. The dipolar oscillations are clearly resolved, and the theory once again matches

the overall shape of the experimental data with a discrepancy for the intermolecular signal captured by the fit factor $q_B = 4.2(3)$. However, this discrepancy does not appear for the intramolecular signal, where the fit factor of $q_F = 1.0(4)$ confirms both that the sample temperature is 5 K $\pm$ 0.1 K and that the theoretical expression in Eq. (6) is correct. As the temperature is raised to 40 K and the polarization value drops significantly, the out-of-phase component disappears, consistent with theory.

Together, the experiments reveal that the theory predicts the out-of-phase component of the intramolecular signal quanti-

tatively, but underpredicts it for the intermolecular contribution. Experiments also show a very slight polarization-dependent asymmetry in the intermolecular signal that is not captured by theory.





A range of experimental and theoretical factors were taken into consideration to potentially explain the origin of the observed discrepancy, the results of which are provided in the SI. Many instrumental factors can be excluded based on control experiments and on the data from the biradical. Pump–probe pulse excitation band overlap for the experimental conditions used

(observer frequency 180.000 GHz, pump frequency 180.050 GHz, pulse widths 40–80 ns) is no more than approximately 2.7% (Fig. S3). This overlap is too small to create significant shifts in the data. Gain imbalance between in-phase and out-of-phase detectors was determined insignificant, as the experiment run with the detector phase rotated by 90 degrees yielded no visible changes in the data (Fig. S4). Experiments were run with detection both on and off the echo to verify that the observed signal arises from the echo and not background of the resonator (Fig. S5). The detector phase was shown to drift slowly over the

course of the experiments. This was accounted for by running the traces with as short of an acquisition time as possible and phasing each trace individually. However, it appears that this is not the origin of the observed discrepancy as traces with backwards and forward sweeps of $t$ give approximately the same result (Fig. S6). Temperature drifts were found to be insignificant, as an temperature sensor near the resonator indicated temperature stability within $\pm 0.1$ K during the experiment. The fact that theory fits the out-of-phase component of the intramolecular signal in Fig. 5b well excludes many potential instrumental

origins of the discrepancy between theory and the intermolecular signal.

Samples were run at sufficient cryoprotectant levels to eliminate aggregation as a source of error, at least for the experiments utilizing a monoradical sample.Canarie et al. (2020) For all 180 GHz data samples were not frozen rapidly outside the spectrometer, but rather after being placed into resonator. For the biradical data, field sweeps of a sample frozen outside and one frozen in the resonator showed little difference (Fig. S8), further indicating that aggregation is not likely a major source of

error.

Possible saturation effects were tested by running a saturation recovery experiment showing that a 500 ms repetition time is sufficiently slow as to not saturate the A-spins (Fig. S7). One interesting possibility could be the occurrence of selective saturation. If the saturation is selective with respect to the A-spin sub-populations, then if the less populated A-spin sub-population is saturated more, the echo phase shift would increase.

The possibility of spectral diffusion of the A-spins can be excluded from consideration as the cause of the discrepancy as it would decrease the overall signal, equally in both the in-phase and out-of-phase signals. Spectral diffusion of the B-spins is expected to have no effect on DEER. One possibility is that there is selective spectral diffusion, affecting the two sub-populations of A-spins differently, though this is unlikely given the excitation profiles of the pulses shown in Fig. (S3).

The discrepancy may be due to flip–flop terms of the A–B or B-–B spin coupling Hamiltonians that are neglected in the

current theory. Inclusion of the A–B flip-flop term has been found to be important for the correct description of DEER signals from Gd(III)–Gd(III) spin pairs with short distances (Dalaloyan et al., 2015; Manukovsky et al., 2017). For a spin concentration $c = 0.1$ mM $= 6 \cdot 10^{16}$ cm$^{-3}$ (obtained from 1 mM total concentration and 10% excitation), the modal nearest-neighbor distance between spins is about 13 nm (Fig. S2), which corresponds to a dipolar coupling frequency of less than 0.05 MHz. The flip-flop term only matters if the dipolar coupling frequency between two adjacent spins is of the same order, or larger, than

their resonance frequency difference $\Delta\omega$. Since A and B spins are frequency-separated by at least 50 MHz (see Materials & Methods), the A–B flip-flop terms are negligible. In addition, the accurate model fit of the intra-molecular signal in Fig. 5(b)





(with $q_F = 1$) confirms that A–B flip-flop terms are irrelevant. Since in the G-band experiments the excitation bandwidth of the pump pulse is about $1.7/t_P \approx 30$ MHz, two adjacent B spins have a typical $\Delta\omega$ on the order of a few MHz. Therefore, the B–B flip–flop terms are likely of minor relevance, although it is not certain that they are negligible.

It is possible that demagnetization effects play a role in the observed discrepancy. In samples with high bulk concentrations of polarized spins, the resonance frequencies of individual spins are shifted due to the presence of a non-negligible dipolar field due to spins that are not excited by either the observer or the pump pulses. This field leads to frequency shifts (Edzes, 1990; Marion et al., 2007). Frequency shifts up to 11 MHz have been observed in a crystal of the organic radical 1,3-bisdiphenylene-2-phenylallyl with EPR at 240 GHz and 8.56 T (Wilson et al., 2020). In our samples, the concentration of these spins (which might be termed C-spins) is about 1 mM, so the demagnetization field and any associated effects are likely negligible.

## 5   Conclusions

Low temperatures and high fields lead to substantial thermal spin polarization. We showed theoretically that this leads to a hitherto unappreciated and unobserved additional phase factor in the DEER signal from a background of a uniform three-dimensional distribution of spins, resulting in a non-zero out-of-phase signal whose amplitude is proportional to the spin polarization. Experimental data, over a range of thermal spin polarizations, i.e. at several temperatures and magnetic fields, for both monoradical and biradical samples confirm the theoretical model describing this polarization-dependent phase factor.

The theory quantitatively matches the experiments for the intra-molecular DEER signal from a biradical, the observed amplitude of the out-of-phase component of the inter-molecular signal contribution is more intense than predicted by current theory. The origin of this discrepancy is unclear. One possibility is that B–B flip-flop terms are responsible. To test this experimentally, spin concentration must be lowered. However, it is unfeasible with current instruments to collect data at much lower concentrations than those shown here, since this would also necessarily require much longer trace lengths to capture a majority of the signal decay. Beyond the reduced number of spins, the signal-to-noise ratio would be additionally degraded by decoherence. Extending the model beyond the dilute limit to a fully coupled spin network to examine the impact of B–B spin coupling would be the next step in narrowing down the origin of the discrepancy.

The polarization phase factor is negligible for DEER experiments performed at 50–80 K and 0.35–1.25 T, temperatures and fields typically used for DEER experiments of nitroxide-labeled proteins, since the spin polarization is at most 1.6% under these conditions. However, the results are potentially relevant for measurements at lower temperatures and higher fields (such as for Gd spin labels measured at 10 K and W-band) and for situations with strong non-thermal polarization such as photo-induced excited-state triplets and optically pumped ground-state triplets.

*Data availability.*   The EPR data used in the main text and supplement are available at https://doi.org/10.5281/zenodo.6537282.



*Author contributions.* SRS, TFP, and SS designed the research. SRS and VPD performed all experiments. SRS analyzed the data. SRS, LM, and SS developed the theory. SRS and SS wrote the manuscript.

*Competing interests.* TFP and SS are members of the editorial board of Magnetic Resonance. The peer-review process was guided by an independent editor, and the authors have also no other competing interests to declare.

*Acknowledgements.* This work was supported by the National Science Foundation (grants CHE-1452967 and CHE-2154302 to S.S.) and the National Institutes of Health (grants R01 GM125753 and S10 OD021557 to S.S.). We thank Thomas Stoll (University of Lorraine, France) for advice on handling the divergence in the sin integral (see Supporting Information).



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
