# Peer review of "The effect of spin polarization in DEER spectroscopy"

_Magnetic Resonance, 2022_

## Community Comment (CC2)

[supplement omitted: unrelated document]

---

## Author Comment (AC3)

**Uniform B-spin distribution**

[Figure]

[Figure]

**Aggregated B-spin distribution**

---

## Author Response (AR1)

**Open discussion comments & responses**

**1 Referee 1**

**1.1 Referee comment**

I can only comment on the theory and simulation side of the manuscript. This is a beautiful work that can be published in the present form subject to the following minor fixes:

1. Right hand side is missing in Eq 13.

2. Brackets are mis-set in Eq S8.

3. Main text talks about modelling the 4-pulse DEER, but the SI then reverts to 3-pulse DEER - wording should be fixed.

4. The free electron g-factor approximation should be declared and explained.

I actually like Sections S1-S3 more than the main text. Relocate those to main text? MR is not some glossy magazine, the readership can handle an equation or two.

**1.2 Authors' response & modifications**

Thank you for the kind words and helpful feedback. Addressing your specific suggestions:

1. $\text{FT}(V_{\text{inter}}(t), \omega)$ has been added to the right hand side of Eq. 13 for transparency.

2. The erroneous extra $\beta$ term has been removed and the parentheses placement corrected.

3. A sentence was added to Line 44 in the main text to clarify that the derivation in S1 is the 3-pulse signal while the main text will focus on 4-pulse DEER model as the results are identical.

4. A sentence was added to Line 64 to indicate the appropriate g-value can be used when necessary but for the theoretical derivation $g_e$ is utilized. The specific $g$ value should not impact analysis as the data are fit for $k$ which encompasses the dipolar constant.

Regarding the movement of equations from the SI text, we have chosen not to move these sections to the main text. Our criterion for separating these sections is so that the focus of the main text is novel physical insight. Any mathematics that are known or not directly relevant to the physical insight were placed in the supplement. They are of course still accessible for those interested. We believe this organization of the content makes the main article accessible to a broad magnetic-resonance audience that may not not be deeply experienced with mathematical aspects of magnetic resonance.

**2 Referee 2**

**2.1 Referee comment**

General comments:

The article "The effect of spin polarization in DEER spectroscopy" by Sweger et al. describes what the title promises. The effect of polarization is usually neglected in the analysis of DEER data, and this paper, en passant, confirms the validity of this assumption for typical measurements (Nitroxides, Q-band, 50 K). Nevertheless, they show that the effect becomes important at high-fields and low enough temperatures. An analytical expression of the intermolecular/background contribution is derived, showing that it also contains an out-of-phase contribution. The analytical expression is confirmed by Monte-Carlo simulations (in the SI), and its functional form is also confirmed experimentally – up to a phenomenological scaling factor. The origin of this factor could not be determined yet, but many possibilities are excluded in the SI, and because the intramolecular contribution fits the theory well.

The work addresses a relevant scientific question and certainly fits the scope of MR very well. It is of course unfortunate that the remaining discrepancy in theory vs. experiment could not be resolved, but I think this is clearly and fairly described, and significant effort was taken to resolve it. This is a common encounter in science and should not be a reason not to publish the work.

The paper is clearly written, and the figures are clear, informative, and also aesthetic. The theory is laid out in simple terms where possible (e.g. fig. 2), and (very!) detailed derivations are given in the SI for the more involved parts. The experiments and data analysis procedures are well described.

The paper shows some overlap with a previous publication (Marko,2013), but – if I see it correctly – arrives at another expression for the intermolecular/background contribution (Eq. (39) in (Marko, 2013) vs. Eq. (8) in the present paper). It is unclear to me what was missing in the earlier work, and a comment in this regard might be helpful. Was something overlooked? Was there another assumption in the derivation?

Specific comments:

1. In Figure 1b, a curve could also be shown for the commonly available W-band.

2. S4 shows two traces, recorded with the receiver phase shifted by 90 degrees. It is a bit surprising that the decay is slightly faster for one of the two. Do you have any idea why this could be? Is it also seen in the absolute value?

3. Line 126: "The measurements were conducted at the field values that gave the maximum echo amplitudes." I was under the impression that in Q-band, one usually applies the pump pulse rather than the observer frequency on the maximum of the EPR spectrum. Is this what you meant?

4. I think the outlook could profit from some short comments about the high-spin case (Gd-Gd, Gd-NO/Trityl. Is the extension obvious or complicated?)

5. The background of single-frequency techniques behaves quite differently. Do you expect polarization effects as well in the case of SIFTER/DQC/RIDME/etc. ?

6. Just an idea regarding the remaining discrepancy: The value for k contains the pump efficiency. If I calculated correctly, at least your Q-band tubes are filled quite high. I don't know your resonator, but I would expect the microwave inhomogeneity to be quite pronounced (not sure about the G-band setup). Even the frequency-swept pulse would lead to a distribution of inversion efficiencies - and thus to a distribution of k. I wonder if this distribution could affect the in-phase and out-of-phase and the inter- and intra-molecular contributions differently? This could be tested computationally quite quickly I think, by assuming some reasonable distribution in microwave amplitudes.

7. I agree with reviewer 1 that you could move some of the theory of the SI into the main text, especially regarding derivations that are new compared to (Marko,2013) but it is mostly a matter of taste.

**2.2 Authors' response & modifications**

We appreciate the detailed summary and feedback of your review.

In response to your last general comment, the results of the integration of Eq. 37 in (Marko, 2013) and S37 & S38 in the present paper disagree. Both papers make the same set of assumptions, however, (Marko, 2013) derives zero for the integration, whereas we found that, analytically and numerically, the result is actually non-zero. Our work therefore corrects an error in (Marko, 2013).

Our responses concerning the specific comments are as follows:

1. A curve for the polarization with respect to temperature for W-band frequency has been added to Figure 1b.

2. This is likely a result of limitations in the range of the phase shifter for the G-band setup. To achieve the 90° change, in some cases, the probe had to be screwed in more tightly (slightly changing the path length to the resonator) leading to a slight change in the pump pulse efficiency and consequently to a slight change of the decay rate, $k = 0.046(3)$ µs$^{-1}$ for 0° shift and $k = 0.053(3)$ µs$^{-1}$ for 90° shift.

3. Line 126 has been modified to clarify the wording to indicate that for data at both temperatures, the field value was chosen based upon the maximum intensity point of a field sweep at the pump frequency.

4. A sentence has been added to the main text (Line 253) to address the relevance and extension of the theory to high-spin systems. Similarly, a line was added to S1 (below Eq. S29) indicating where in the theory would differ in samples that are not thermally polarized.

5. We would expect there to be effects from polarization on any of the mentioned experiments, however, the background theories are quite different from the work presented in this manuscript. Unfortunately, there is no way to comment on the extent of those effects without doing a detailed theoretical analysis which is beyond the scope of this work.

6. The model would hold in the case of inhomogeneity in $k$. As a test example we examine a fairly broad Gaussian distribution of $k$ with mean of 1 µs$^{-1}$ and a full width at half maximum of 0.6 µs$^{-1}$, as shown in the top panel of the plot below. The decay curves calculated by integrating over this $k$ distribution and calculated for the central $k$ value are nearly identical (bottom panel). This is valid for both the in-phase and the out-of-phase components. In particular, the amplitude of the out-of-phase component is barely affected. Therefore, we conclude that $k$ inhomogeneity is likely not responsible for the observed mismatch between theory and experiment.

[Figure]

7. We have chosen not to move these sections to the main text. Our criterion for separating these sections is so that the focus of the main text is novel physical insight. Any mathematics that are known or not directly relevant to the physical insight were placed in the supplement. They are of course still accessible for those interested. We believe this organization of the content makes the main article accessible to a broad magnetic-resonance audience that may not not be deeply experienced with mathematical aspects of magnetic resonance.

**3 Community 1**

**3.1 Comment**

This an interesting paper improving understanding of physical processes and interpretation of DEER measurements, which are often attempted to use for resolving important questions of structural biology. This paper goes into the analysis of the situation when DEER experiments are performed at low temperature and high magnetic fields, that is, when a significant spin polarization is present. It predicts an out-of-phase signal component of the inter-molecular contribution, which was overseen and was not analyzed in the previous work. The predicted out-of-phase inter-molecular component is used to explain qualitatively experimental data, although there is still quantitative mismatch between theory and experiment. The magnitude of the out-of-phase inter-molecular component is determined by the value of the integral (S58), the convergence of which is not straight forward and apparently, this integral relay strongly on the assumption of homogeneous and isotropic distribution of spins on long distances R in the sample. I think, some inhomogeneity or anisotropy of spin distribution can change the value of this integral significantly, perhaps much more than of the corresponding in-phase component integral. However, these are my rough estimation. Probably, some more extensive investigation/comments regarding this point would be useful in the paper or in the discussion. Anyway, apparition for an interesting work!

**3.2 Authors' response & modifications**

Thank you for the interesting insight into the work. Regarding the impact of inhomogeneity on the integral, we do not believe it would change the result. The plots below show the results of Monte Carlo simulations in which the B-spins are distributed uniformly (top row), as the analytical theory assumes, or aggregated (bottom row). The result does not show a change in the presence, shape, or magnitude of the out-of-phase component and only changes the rate of decay, $k = 0.5 \ \mu s^{-1}$ for the uniform distribution and $k = 1.85 \ \mu s^{-1}$ for the aggregated distribution. As the decay rate is fit for each experimental dataset shown in the main text and the experimental conditions are such that aggregation should not be an issue, we do not expect that inhomogeneity is a concern and therefore have only addressed it briefly in the main text and supplement.

**4 Community 2**

**4.1 Comment**

The paper is important contribution in the field of dipole spectroscopy, it provides rather clear picture of the spin dynamics effects.

I think it would be suitable to cite our early theoretical paper (attached) Maryasov A.G., Dzuba S.A., Salikhov K.M. Spin polarization effects on the phase relaxation induced by dipole-dipole interactions. J. Magn. Reson. 50: 432-450, 1982. There we demonstrated that out-of-phase component will appear in the case of finite value of $\epsilon(B,T)$ given in Eq.(5) of the manuscript, see Eq(29) of the old paper. We studied dipole line broadening that time, and spectral diffusion caused by d-d interaction in 3D system with randomly distributed PCs, so didn't concentrate on out-of-phase signal. Now those results may be treated as background signal calculations. High Botzmann polarization of spins in pair of radicals leads to the triplet character of the pair initial state regardless of the strenght of dipole spin coupling, equlibrium density matrix is proportional to S1z+S2z+cS1zS2z, where c is a non-zero constant even when d-d interaction is neglidgeble. High Boltzmann polarization also reduces spectral diffusion effects in weakly coupled systems. It seems to me that our old paper is relevant to be cited.

**4.2 Authors' response & modifications**

Thank you for the response and your interest in the present work. We have added a citation at Line 44 to recognize the contributions of your work to the theory presented in this manuscript.

**Uniform B-spin distribution**

[Figure]

[Figure]

**Aggregated B-spin distribution**